# The Impact of Different Overlay Materials on the Tactile Detection of Virtual Straight Lines

**Patrick Coe \*** , **Grigori Evreinov** and **Roope Raisamo**

TAUCHI Research Center, Tampere University, 33100 Tampere, Finland
\* Correspondence: patrick.coe@tuni.fi

**Abstract:** To improve the perception of haptic feedback, materials and sense-modifier effects should be examined. Teflon, Nylon mesh, and Silicone overlays were tested in combination with lateral vibrations to study their impact on the tactile sense. A feelable point moving along a line was implemented through the use of a dynamically moving interference maximum generated via the offset actuation of four haptic exciters affixed to corners of a Gorilla Glass surface. This feedback was presented to eight participants in a series of randomized experiments. Both the Nylon mesh and Teflon covering revealed a statistically significant ($p < 0.05$) impact of improvement to the user performance in the task of dynamic haptic virtual straight lines localization. While Silicone covering, having three times greater friction than Gorilla Glass, has less or no impact on both decision time, the number of task repetitions, and error rate ($p > 0.05$). The lateral vibration modifier (60 Hz) can also successfully be used with an increase in performance by about twofold, at least that was demonstrated for both the Nylon mesh and Teflon covering.

**Keywords:** communication hardware; interfaces; actuators; human-centered computing; human–computer interaction; interaction devices; haptic devices

## 1. Introduction

Multi-finger pad stimulation can be achieved by creating a local interference maximum (LIM) moving dynamically across the touch surface of an electronic device that is in contact with a user's skin. As the precision of LIMs continuously improve, we may begin to see advancements in the technology, for example, to display tactile graphical images in lieu of expensive Braille displays [1,2]. Furthermore, the local perception of constructive wave interference vibration signals can be enhanced by a superimposed frame of reference (FoR) modulating signal [3]. It is already well-known that intermediate materials can enhance the sense of touch. There exists, for example, a patent for a touch-enhancing pad [4] that consists of a lubricant placed between thin plastic sheets. Auto body shops commonly use a similar technique involving the use of cellophane film to enhance the sense of touch when examining the finish on vehicles [5].

There is an ongoing demand to identify the materials and methods capable of enhancing tactile perception in combination with the specific signals and parameters to improve the localization of vibration and complex haptic signals [6–9]. Furthermore, there is a need for materials that can be, for example, integrated with protective components or applied to device surfaces as a more effective solution to further enhance tactile perception [10–14]. At the moment, enhancing tactile sensitivity in this manner worsens the signal-to-noise ratio. Existing successful experimental haptic techniques rely on the use of closed-loop feedback to achieve high-fidelity signal localization, raising significant difficulties and limitations for practical implementations in mobile devices.

A new method that can enhance human tactile perception is the use of constructive wave interference vibration signals. This has been investigated by Kurita through the use of a wearable device [8]. In this case, the wearable device includes a set of haptic

exciters that create the local interference maximum and an additional haptic actuator to provide sensorimotor enhancement. The sensorimotor enhancement is achieved through the introduction of white-noise vibration, known as the Stochastic resonance effect [15]. This successfully improves perceived feedback cues, leading to improved precise dexterous manipulation. Kurita, with his research team, found that by introducing this white-noise vibration to the grips of forceps, they could enhance a surgeon's sensorimotor abilities [16].

It is supposed that the superimposed texture of different materials usually used to protect a touch surface can behave as a FoR by modulating the perceived haptic signals of a dynamically moving LIM. This is achieved by deviating the FoR textured touch surface or a contact surface covered with a material having a specific micro-structure (Figure 1) with a given vibration frequency and magnitude. Such a technique could help to achieve high-fidelity imaging in 2D haptic space without the constrictive signal demands of existing haptic actuator technology.

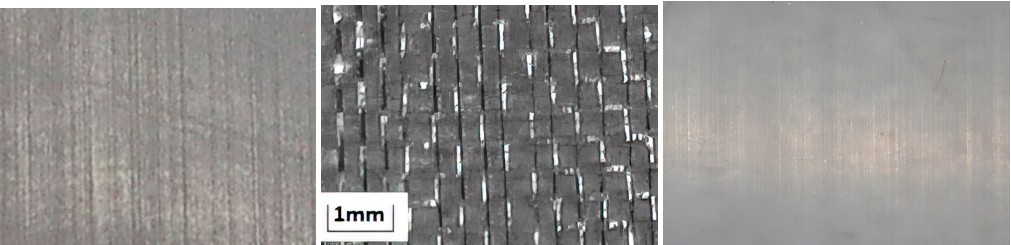

**Figure 1.** Surface structure of Teflon (**left**), Nylon mesh (**middle**), and Silicone (**right**).

Previous works have studied methods of achieving LIM over nonstandard form factors [17] and different mediation mediums [18]. Similar methods in use are also detailed by Huawei in a recent patent [19]. While previous research has aimed to make technical improvements to the physical displacement at the point of the LIM, this current research attempts to study ways in which the perception of the LIM can be improved. This will also test the perceptibility of previously achieved LIMs that have been primarily studied via the collection of surface data with the laser Doppler vibrometer.

The study seeks to better understand the behavior of the well-documented and widely available materials of Teflon, Nylon, and Silicone in relation to haptic feedback and perception as follows [20–23]. These materials are already in widespread use in consumer products [24–26]. Therefore, we aim to find out if any of these materials can enhance both the perception of localized vibrotactile feedback as well as enhance the feedback itself. While there have been strides in the creation of localized vibrotactile feedback that occurs away from the source actuator, we find that many of the published studies result in localized feedback, which is only slightly perceivable from subsequent global vibrations over a device. Keeping in mind that these studies are performed in controlled laboratory settings, we believe that the studies of materials for haptic enhancement can improve these results. Thus moving towards the goal of localized haptics integrated into consumer devices used by a wide and active demographic of users with varying needs.

This paper's structure following the introduction is as follows: We will begin by detailing the hypothesis before going into the background and related works as they pertain to the current work. We will familiarize the reader with the experimental design, which goes over the materials and parts used for the construction of the device in use. Here we will also review the doppler vibrometer data to get an understanding of how modifier materials affect surface vibrations. After this, we will present the methods, which primarily include the experimental procedure. This is immediately followed by the results, after which we present a discussion section that highlights the key findings in the data we have collected. Lastly, we present a brief conclusion.

*Hypotheses*

The hypotheses for this experiment were the following:

**Hypothesis 1:** *The human sense modifier material will impact human performance on the exploration of virtual straight lines by decreasing the error rate and decision time as well as the number of repetitions (test trials) each virtual line (haptic object) is presented for.*

**Hypothesis 2:** *The local vibration modifier (as a sensorimotor enhancer [8]) will impact human performance by decreasing the error rate and decision time as well as the number of repetitions of each virtual line (haptic object) presented.*

## 2. Background and Related Works

While extensive studies have been conducted by many researchers on the subject of physical vibration localization, there is a need to better understand how the cutaneous sense of virtual haptic objects and the local vibration cues can further be enhanced. For this, we should discuss some of the previous work conducted in this area, some fundamental properties of skin, as well as details of the materials in use.

The ability to create localized haptic signals is well-documented by researchers. Previous studies [18] have shown success in the ability to create small, localized regions of increased normal displacement through the use of offset actuation of voice coil actuators. This work is a resource of new digital signal processing techniques for the express purpose of generating a LIM on a variety of surfaces, including nonstandard curved ones [17].

Existing research shows that the friction behavior of human skin is correlated to the material and surface properties of the skin in connection with the contact material as well as the presence of substances such as water, sweat, or skin surface lipids. Micro-ridges of the skin under the fingers also affect tactile stimulation in correlation with the micro-extrusions of the contact surface [14,27].

Both the micro-ridges and the micro-extrusions interlock to generate optimum tactility while interacting with the given surface. The viscoelastic material properties of the skin and its micro-ridges give it the property of a soft elastomer [21,28]. Therefore, we believe that utilizing different textured materials with varied micro-extrusions will create the simulation of dynamic rendering in a localized area on the flat Gorilla Glass surface.

For this study, we selected three overlay materials to place over the existing Gorilla Glass surface of our tablet display. This includes a Teflon sheet, Nylon mesh, and a soft Silicone liner. The materials selected for use during this experiment were chosen due to their low manufacturing cost, physical properties as they relate to touch perception, as well as their current existence as materials used in current mobile phone accessories and cases. We selected these three criteria with the intention of providing data relative to materials that could provide a realistic implementation of haptic mediation techniques in current consumer devices.

## 3. Experimental Design

The experimental tactile interface has been carefully designed for ease of use, familiarity, and durability. The chosen device to implement the testbed of vibrotactile localization is of the Microsoft Surface Go tablet. This tablet was chosen such as to provide a familiar form factor for the participants to explore. The use of a tablet device also allows us to integrate visual feedback for use in future experiments.

In Figure 2, top, we can see the display surface of the tablet. In each corner, a Tectonics TEAX1402-8 actuator is placed. Tectonic actuators are used due to success in previous related work [29], where striking the glass surface is necessary. The properties of Gorilla Glass make adherence of many adhesives difficult. We found that epoxy glue works well as an adhesive intermediary. A layer of double-sided tape is used to adhere the actuator to the epoxy intermediate. Actuation offsets for localization LIM peaks are determined by cycling through a range of given offsets between the actuators to find the offsets that provide the highest peak vibration at a given location. This is the same method as described in detail in other previous work [17,18,29].

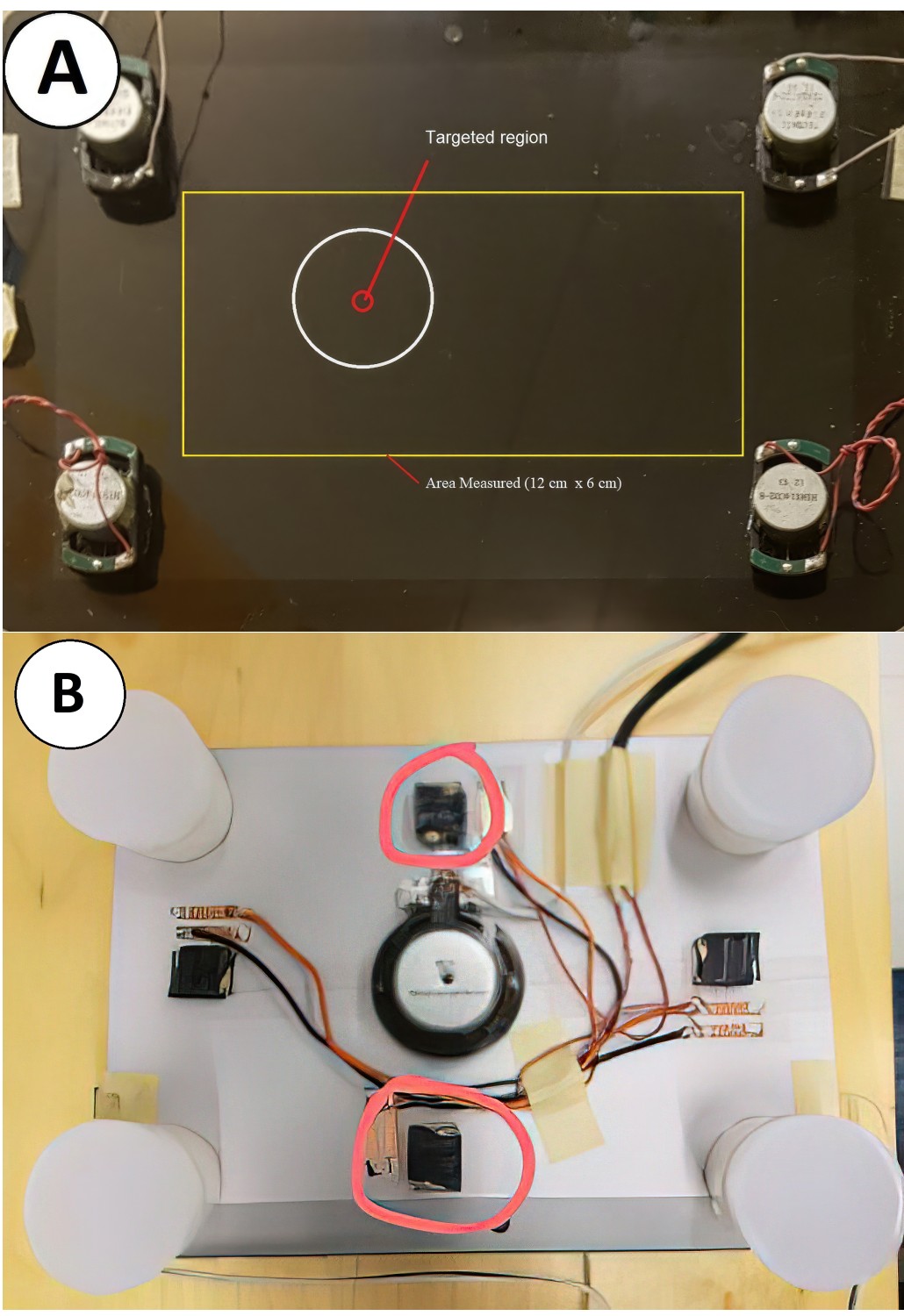

**Figure 2.** (**A**) View of the top of tablet display surface used, the Surface Go. Four tectonic actuators can be seen placed at each corner. (**B**) L5 actuators can be seen placed along the ends of the *X* and *Y*-axes of the tablet (circled).

Knowing these offsets, we then program points of localized feedback to occur in rapid succession from near the center of the tablet, moving toward the edge of the tablet following a straight line. During this experiment, we use ten evenly distributed localization points along each of the five lines to create the illusion of a vibration point that adheres to the given virtual line. The speed at which the point moves is limited by the offsets required to produce each given point. Because different offsets are required to target points on each

line, the speed at which the point moves across a line will not be identical from line to line. That said, we have chosen ten as the quantity of given localization points as that achieves a single motion from end-to-end of the line within 0.5 s or less.

Underneath, on the back of the tablet, there is another distinct set of actuators. Four next-generation Lofelt L5 actuators [30] are placed to provide feedback in the $x$ and $y$ directions. In this experiment, only the Lofelt actuators placed along the $Y$-axis are triggered. A frequency of 60 Hz has been chosen for lateral vibration as it is the resonance frequency of the Lofelt L5 actuator. The purpose is to understand if the 60 Hz lateral vibration can effectively induce an undulation sensation with respect to the FoR, thus resulting in an increase in touch sensitivity. Therefore, by aiding a participant's cutaneous ability to easily feel the targeted, localized signal dynamically moving along a particular direction on the surface of the tablet display. The average duration of each point is near 50 ms, with variation depending on the location of each line. With the 60 Hz lateral vibration being equal to 16.7 ms per single actuation, or three times per point.

An Arduino Due has been selected to manage the control signals for all mentioned actuators. Output signals are running via wire to external L298 motor controllers, which drive the actuators. Silicone rubber legs have been molded and attached to the underside of the tablet. This is carried out to provide vibration isolation, muffling vibrations from the tablet to external surfaces (table) that may distract participants during testing.

### 3.1. Overlay Material as Touch Modifier

Surface material properties can affect the way a given surface is perceived [20,22,23,31]. Materials both modify how localization occurs, subjective user perception of haptic signals as well as virtual haptic objects. Improving the accuracy of sensing directions of virtual straight lines presented dynamically through tactile signals moving over the screen surface, as well as differences in how pleasant or frustrating the exploration of the virtual haptic object on a surface might be. As the reference baseline material, we used the Gorilla Glass surface of the Microsoft Surface Go tablet. Gorilla Glass has a coefficient of friction of $0.2 \pm 0.09$ [32].

For this experiment, we have chosen to use Teflon, Silicone, and Nylon mesh as overlay materials (Figure 1). The Nylon mesh in use is 71 microns thick with 65-micron hole openings and features excellent strength and low elongation properties. While Nylon mesh has the lowest friction ($0.37 \pm 0.09$) [20,21,23], it provides a comfortable textured feel to the hand and is widely used for 3D printing. The Teflon sheet we used is approximately 100 microns thick. It is not significantly different from Nylon mesh but devoid of texture (friction is about $0.43 \pm 0.09$), having a comfortable, smooth feel and very high temperature resistance [20,22]. Silicone soft liner material is often used for a prosthesis, has the highest coefficient of friction ($0.61 \pm 0.21$) against human skin [23], and is widely used as a material for smartphone protective cases.

As shown in Figure 3, materials are placed over a Mirka ECOWET brand silicon carbide-based P800 grit sandpaper-backed sheet, which is used to adhere to the surface of the tablet. The cross-section, as shown, has been designed for the practical purpose of running the test on participants. Materials placed directly on the Gorilla Glass slide around and do not stay flush with the surface, making it difficult to explore. Because of the anti-fingerprint oleophobic properties of the Gorilla Glass, it is difficult to find an adhesive that sticks well directly to the surface without permanently damaging it. We found that the sandpaper grit easily stays flush with the glass surface, does not slide around, and is easy to swap out during user testing. To keep materials flush with the surface, they were then adhered with total coverage to the sandpaper backing using 3M 9080HL Double coated tape, a general-purpose thin (0.16 mm) tape that will minimize any alteration to the vibration signal by the use of generally uniform, totally adhered, sub-layers. These combined sheets are then cut into a shape that covers half of the surface where the test is being conducted while leaving space around the placed Tectonic actuators. We found that this combination allows for materials to quickly adhere to the Gorilla Glass surface,

laying entirely flush with the Gorilla Glass surface without allowing lateral movement of the material to occur. This sheet is then clipped on to avoid any accidental lifting of the sheet from the glass. The high grit used also ensures that during testing, the rough surface of the sandpaper is not felt on the side the user is exploring. The user will only feel the surface material when exploring with their fingers.

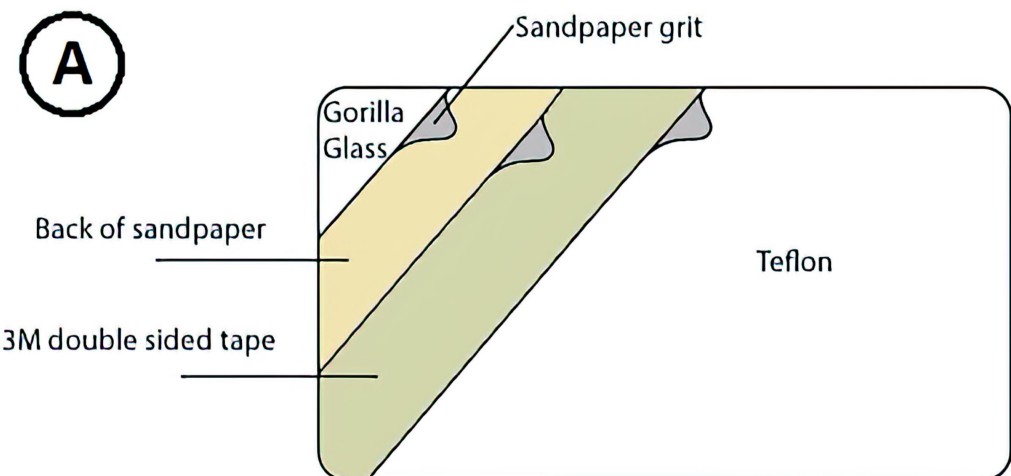

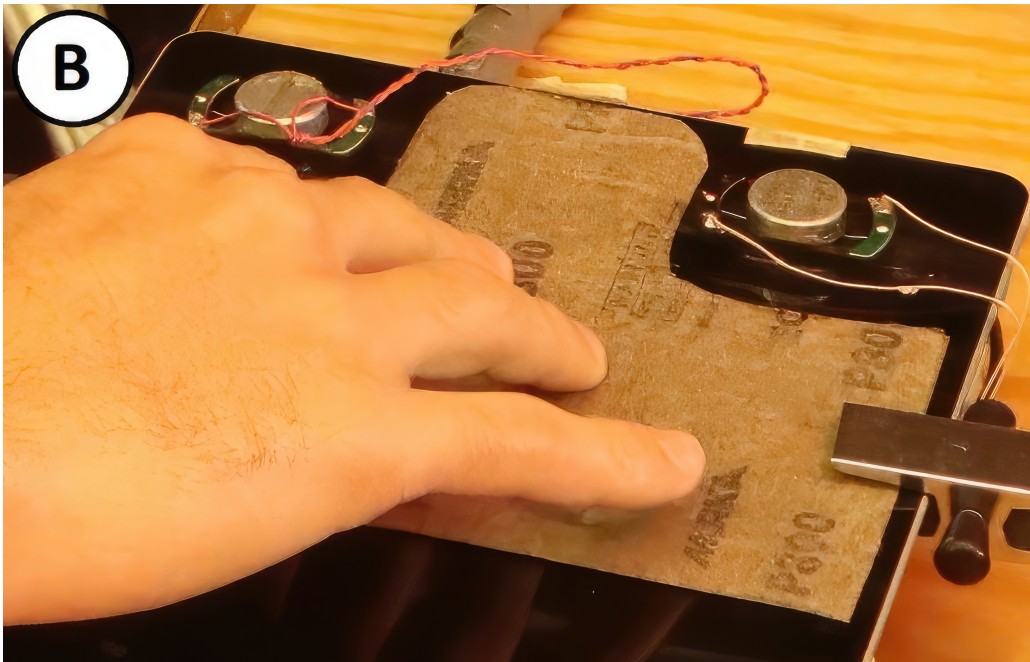

**Figure 3.** (**A**) Cross-section of the layers of the overlay surface. (**B**) Overlay placed on tablet surface with a participant's hand exploring for localized feedback.

### 3.2. Doppler Vibrometer Data

To get a visual understanding of how seismic waves of a targeted interference maximum propagate over a surface when virtual straight lines are presented with different material overlays, we had our display surface measured using a laser Doppler vibrometer. Using the initial materials (Gorilla Glass, Teflon, and Nylon mesh), preliminary information about how the materials modify the propagation and amplification of waves was measured.

Reviewing the data collected with the Doppler vibrometer, we were able to receive an initial glimpse of how vibration localization is modified due to distortion and dissipation at the targeted point of localization. In all cases, a clear maximum displacement peak around

the targeted area was observed, with each material changing the form of the peak vibration presented across the surface of the display. For example, over Gorilla Glass, we could see that the area of impact was visibly targeted to the left. Although the resulting peak was properly localized, we also revealed a wide region with a general smooth diminishing of peak displacement away from the point targeted position. The affected region was quite large, implying a low-resolution localization effect. In the case of the Nylon mesh configuration, we found that the displacement region becomes narrower, yet with smaller, noisier peaks around the targeted localization. This might imply a further localized tactile point, yet it remains to be indicated how the noisier region surrounding the central peak will affect perceptibility. The case of the Teflon configuration demonstrated further precision without the noise previously witnessed with the use of the Nylon mesh material. The quick drop-off in displacement from the point of localization should improve the discernibility between said contact point and the rest of the display.

Overlaid vibrometer data can be seen in Figure 4 with further details which can be reviewed in Figure 5. These are the graphs of the measured maximum displacement achieved over the Microsoft Surface Go tablet. As mentioned earlier, a point on the left-hand side of the tablet was targeted for vibration displacement localization (Figure 2 top). With the Gorilla Glass, the graph shows a smooth but wide curve with a peak centered towards the left of the tablet (Figure 4A), confirming our initial review of the collected data. Our second graph displays the data collected in relation to the Nylon mesh (Figure 4B). It indicates a similar wide displacement with a distinct peak at the targeted localization point. The curve itself is still a bit wide, with peaks and valleys that introduce noise to the point of localization. The final graph of data collected from localization targeting Teflon (Figure 4C) reveals sharper, stronger points of displacement than either Gorilla Glass or the Nylon mesh. The highest point of displacement for this graph coincides with the targeted localization point, with some secondary peaks above and below it (Figure 5). With Gorilla Glass and the Nylon mesh, we measured a normal surface displacement of about $8.3 \times 10^{-6}$ m, while with the Teflon sheet, this was amplified to $12 \times 10^{-6}$ m.

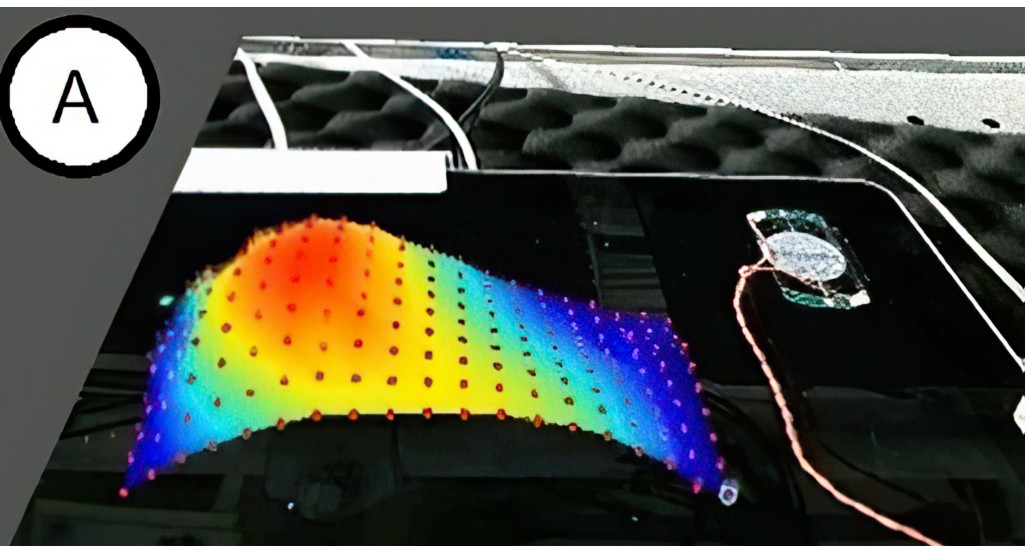

**Figure 4.** *Cont.*

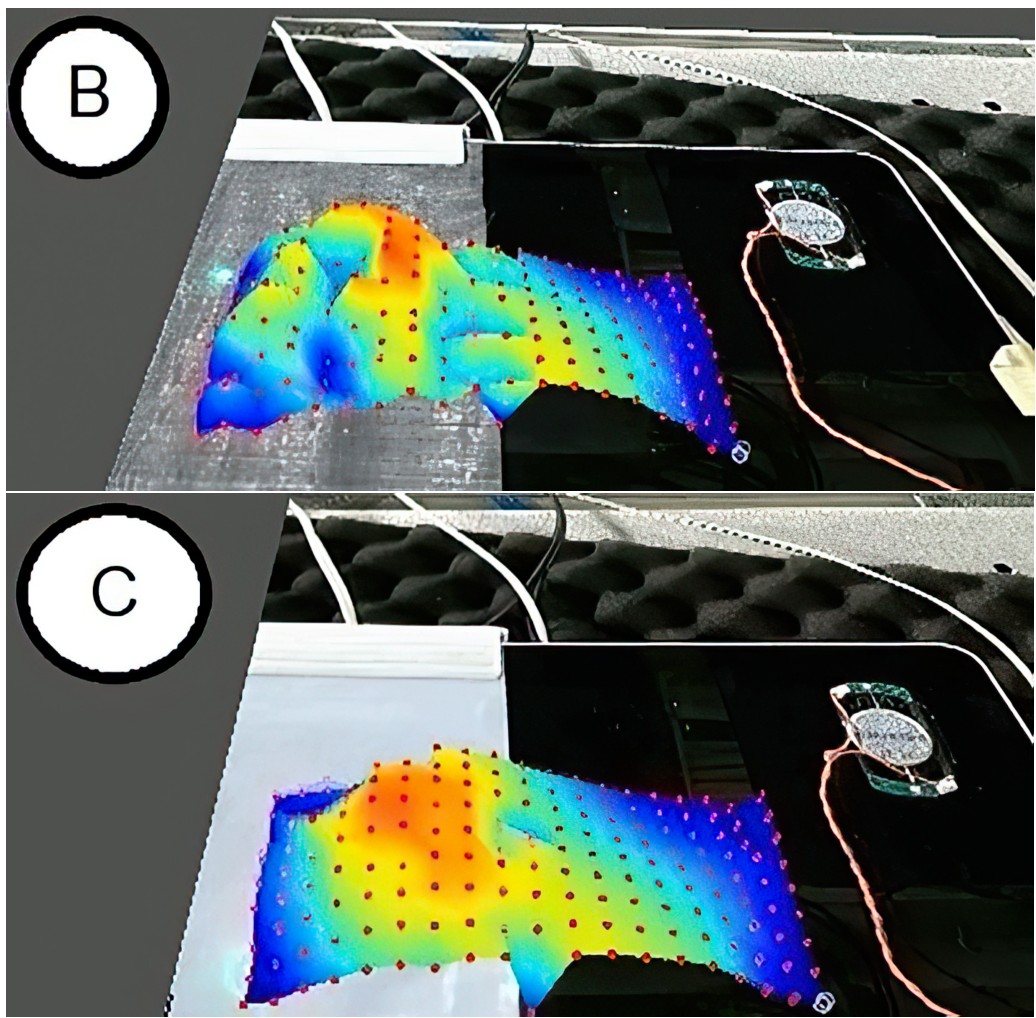

**Figure 4.** Collected vibrometer data overlaid with images of tablet display. From top to bottom: (**A**) Gorilla Glass, (**B**) Nylon mesh, and (**C**) Teflon.

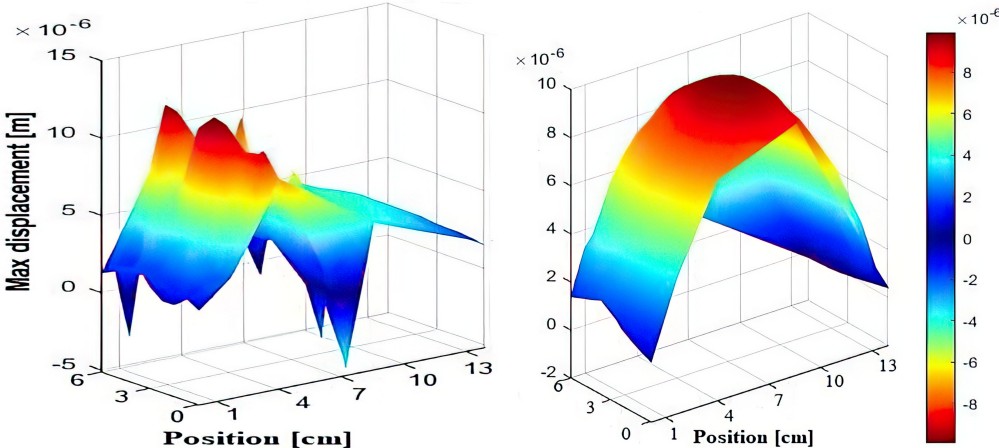

**Figure 5.** Vibrometer displacement data of Teflon (**left**) and Gorilla Glass (**right**).

　　　For all three tests, an identical signal was used, the only change being the overlay material. This data would indicate that Teflon is the most effective at increasing the localization of seismic wave signal interference.

## 4. Methods

*Experimental Procedure*

Eight participants (three female, five male) from ages 28–36 were selected for the experiment. All users were predominantly right-handed. Participants were unfamiliar with the overall subject of haptics. Participants were also unfamiliar with the experiment itself before participating. Each haptic signal presented to a participant imaged a sloped line beginning from the left edge of the display and traveling to the center of the display generated via the use of interference waves. That is to say, a moving pulse along a given line should be felt. In random order, for each material, virtual straight sloped lines were presented six times in each of five directions with either the lateral vibration modifier (LVM) or no LVM present (240 signals to be analyzed in total).

The sloped lines presented were chosen from five angled lines placed equidistantly (Step = 22.5 deg.) from 45 to −45 degrees (Figure 6). Each sloped line was presented 12 times, with half of these presentations being the $Y$-axis 60 Hz LVM signal. Each sloped line was continuously presented until the participant entered a response or a timeout (60 s) was reached before moving to the next random signal. To avoid biased data and discrepancies due to learned behavior, the presentation of the four materials was randomized to each participant. Participants were instructed to explore the surface of the tablet via touch in whichever manner they felt most comfortable.

The configuration of Tectonic actuators produces audible sound when presenting different signals. To prevent this sound from influencing participant responses, all participants wore a pair of 3M PELTOR Optime III H540A with a rated SNR of 35 dB.

During the experiment, participants would input their response through a keypad with what displayed line they believed they were being presented. All participants claimed that they properly understood the task, as well as how to use the keypad to input their response.

After the completion of the experiment, participants were asked to first fill in a task load index questionnaire. This was followed by a general questionnaire related to the experiment.

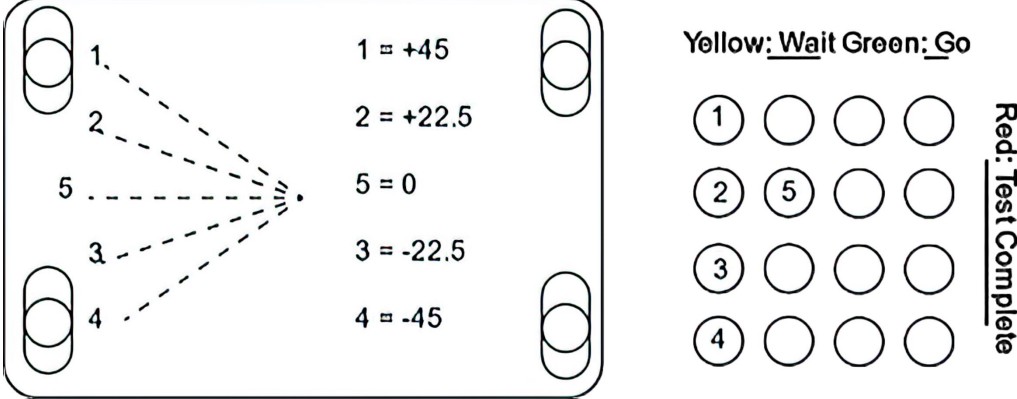

**Figure 6.** A reference diagram available to participants to understand the task. Localized vibration originates from the left edge of the surface, moving toward the center based on a randomly chosen angle. The keypad is represented on the right. Users select a key to be pressed based on what they feel matches with the diagram on the left. Angles on the diagram are shown from 45 to −45 degrees.

## 5. Results

Subjective data from the forms and questionnaires filled in by participants show initial promising results on the impact of different materials on the subjective cutaneous sense of the moving interference maximum signal localization. Objective data collected from the input presses of the participants solidify these findings as well as gives insight into the effectiveness of the LVM. We have found that both the material in use along with the LVM, seem to affect how each participant perceived the displayed dynamic virtual haptic objects presented as straight sloped lines.

## 5.1. Subjective Data

The subjective data were collected immediately after the experiment was over. Participants filled out both a task load index survey as well as a standard questionnaire. The collected results reinforce the collected objective data.

### 5.1.1. Task Load Index

The task load index survey was presented to participants at the end of the experiment in order to keep responses as relevant as possible. Questions were measured on a scale of 0 to 100. A perfect workload was considered one where the question "How successful were you in accomplishing what you were asked to do?" was given a score of 100, and all other questions were given a score of 0.

Figure 7 displays the collected TLX responses overlaid into a radar chart. The overlaid radar chart reveals that some participants felt more at ease with the task than others. We believe that familiarity with the experiment, along with ongoing technical improvements, can lead to a closer to ideal workload.

In relation to how mentally demanding the task was participants responded with an average score of 51 on the TLX scale. This score indicated that the task may, at times, require higher levels of focus than what might make the task difficult yet borderline acceptable. As the task had been presented at the time of the experiment, we do expect a higher level of mental demand from the users.

A score of 39 on the TLX scale was recorded in relation to the physical demand of the experiment. This would indicate that the task in this regard was acceptable and not overly physically demanding. This would allow participants to better focus on exploring the surface with their fingertips.

Participants, in general, did not feel rushed during the experiment, with a clearly acceptable workload score of 19. A high level of confidence was indicated in participants' ability to complete the task, with an average response of 55 on the TLX scale. With a response of 46, participants also indicated a moderate amount of effort was taken to accomplish their level of performance, with a similar accompanying level of insecurity, stress, and annoyance toward the task, with a score of 51.

All in all, we can say that responses to the task load index survey indicate that participants were presented with what can be considered a reasonable task that should have little impact on the participant data collected during the experiment.

### 5.1.2. Questionnaire

Participants were given a questionnaire to fill out after completing the task load index survey. In this questionnaire all participants responded that they felt the moving vibration that occurred happening. All participants reported accurately perceiving the motion of the vibration occurring from the left to the center of the device.

We asked participants to rate the materials they had been presented with on a scale of 1 through 5 based on their own personal opinions of which material they thought both felt better and most accurately perceived localized vibration. As expected, Teflon was considered the best-rated material on this scale at 3.5. The Gorilla Glass surface similarly received a close 3.4 score. The Nylon mesh and Silicone overlays received poorer ratings, 2.9 and 2.8, respectively. This suggests that the smooth, slick surface provided by both Teflon and Gorilla Glass are more desirable to participants than those with increased friction, such as the Nylon mesh or Silicone overlay.

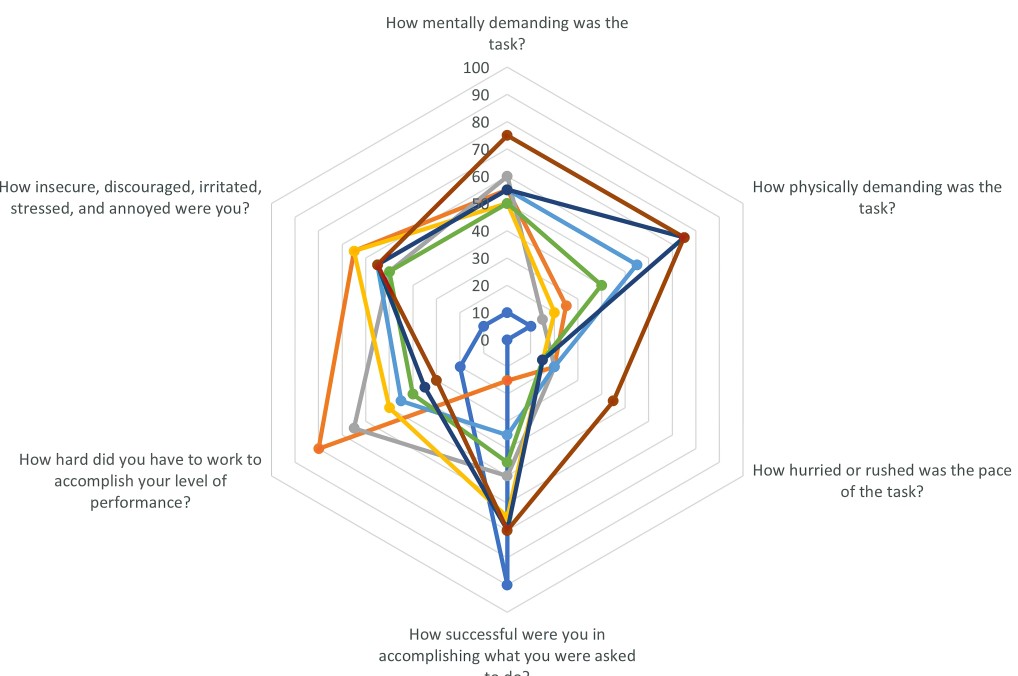

**Figure 7.** Overlaid participant data collected from TLX questionnaire represented in a radar chart.

We then asked the participants about how well they felt the discreet slopes were presented during the experiment. We received mixed opinions on how well participants believed they could feel the discreet localized vibrations. Scored from 1 through 5, we received an average response of 2.75. The wide range of responses indicates that while an individual participant's perception can vary, the signals presented were reportedly perceived by all participants.

We then asked the participants open-ended questions to better understand in what applications they could see this technology used. Several mentioned how it could be used in gamepads to enhance the gaming experience, others mentioned it could help improve ATM displays, and general smartphone and tablet usage was also discussed. Even the concept of AR clothing was mentioned, to either direct a user's movements or help enhance sensations. Increased accessibility for the blind, or for the young and old was mentioned often. The responses shows that the participants quickly understood the practical applications for this technology and were eager to share new related ideas.

Overall, we were satisfied with the data collected from the filled out questionnaires.

*5.2. Objective Data*

We consider the participant's input data during the display of haptic signals to be objectively collected data in reference to user perception.

When reading the top chart in Figure 8, Pressed indicates that a selection was made. The only time a selection would not be made is if output would have repeated until it timed out. Repeats refer to the number of times an output signal was repeated. An output signal is continuously repeated until a selection is made or times out. Therefore, a higher number of repetitions indicates a slower decision time. ErrRate indicates the number of selections that were incorrect.

One of the first findings we notice is the influence of the 60 Hz LVM signal in all of our material tests (Figure 8). Regardless of material, when the 60 Hz signal is introduced, we see an improvement in both response (decision) time and accuracy (error rate). Participants were never informed on whether the 60 Hz signal was activated or not at any time during the test to prevent any potential bias in the evaluation.

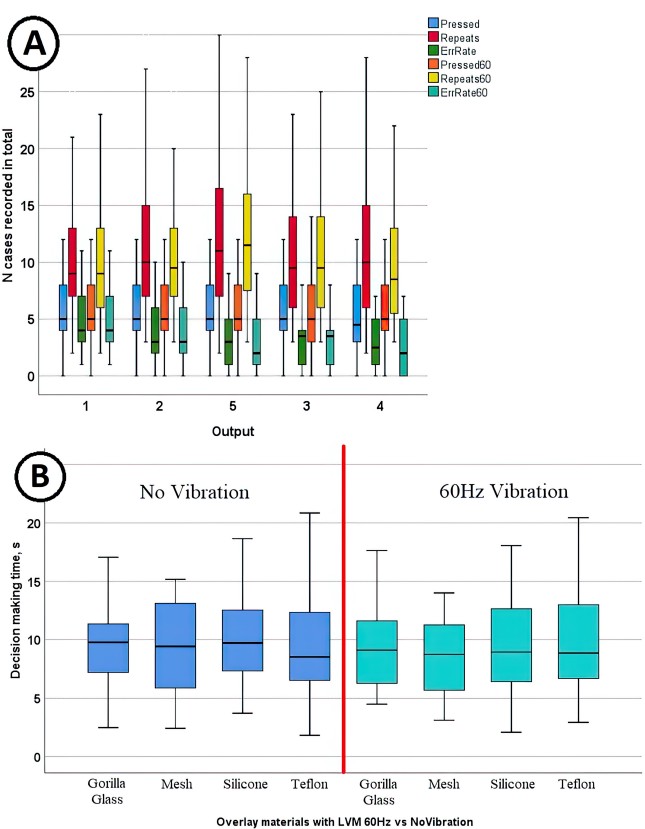

**Figure 8.** (**A**) Comparison on Gorilla Glass between using the 60 Hz modifier and not. Pressed refers to the number of times a selection was made. Repeats refer to the total times an output was repeated. ErrRate indicates incorrect responses. (**B**) Comparison of decision time (s) over materials.

Comparing materials, we revealed that changes in participants' behavior (error rates, decision time, index of assigned direction responses, and the number of trial repetitions) do not significantly vary between materials. What does change is the participant's decision time, with a participant decreasing their decision time when using any of the intermediate overlay materials. The difference in decision time (in ms) is most prominent when comparing the Gorilla Glass surface with that of the Teflon overlay while inducing the 60 Hz signal (Figure 8). Inducing the 60 Hz signal improves decision time for both materials, closing the gap. These collected data are further analyzed via a repeated measures paired samples *t*-test (Table 1), complementing the findings.

**Table 1.** Results of Repeated Measures Paired Samples *t*-test.

| G.Glass MSM | Df = 299 | t (NoLVM/LVM60) | Sig. (2-Tailed) $p < 0.005$ |
|---|---|---|---|
| G.Glass | DTime | 3.14/1.55 | 0.002/0.122 |
|  | ERate | 1.07/1.50 | 0.284/0.133 |
| Nylon | Rep | 3.23/2.17 | 0.001/0.031 |
| G.Glass | DTime | 3.12/3.34 | 0.002/0.001 |
|  | ERate | −1.82/−1.97 | 0.069/0.049 |
| Teflon | Rep | −2.94/−3.79 | 0.004/0.000 |
| G.Glass | DTime | 0.96/1.67 | 0.333/0.094 |
|  | ERate | −0.13/1.10 | 0.895/0.270 |
| Silicone | Rep | 0.95/0.89 | 0.342/0.372 |

## 6. Discussion

Our research has shown that previously achieved interference maximum local vibrations are well-distinguishable by fingertips. Additionally, it has shown that intermediate materials are able to enhance the human cutaneous sense of the moving interference maximum signal. The best material to do so from our empirical findings was that of Teflon; which was also subjectively considered to be the best material to interact with by fingertips. Subjective data coincides with existing research that describes pleasant tactile surfaces as smooth or slippery and less pleasant surfaces as rough or sticky [33,34].

Haptics play a vital role in our visual-centric culture. By introducing high-fidelity haptics into, for example, the classroom, we can encourage the development of perceptual and motor abilities; especially for those with visual impairment. At this stage, research would suggest that learning can be improved at any development stage among the majority of students [35].

The use of modifying sense materials (MSM) was found to decrease error rates, decision time, and the number of repetitions for each direction of playback of the moving interference maximum signal (Table 1). The Pairwise comparisons with Bonferroni correction show that in the case of Teflon as well as the use of the Nylon mesh, we see a significant reduction in the number of repetitions required to make a decision, significantly reducing the decision time itself. Silicone did not show a significant improvement in any area. Teflon was the only material found to achieve a significant decrease in all three areas, meeting the requirements set out in our first hypothesis.

Furthermore, it was shown that a 60 Hz orthogonal vibration could further improve subjective perception. As for our second hypothesis regarding the impact of the 60 Hz LVM, a significant difference when reviewing the collected objective data has not been shown at this time. The 60 Hz vibration was chosen as a starting point (close to the resonance frequency of L5 actuators), but is not necessarily the best frequency to use to enhance perception. Because inducing noise can enhance perception [8,13], it is not possible to make conclusions based on limited data collected from our measurement tools. Instead, a range of frequencies would need to be explored by participants in order to reach a better understanding of how cutaneous perception could be enhanced. This can also be said of the magnitude of the actuation, as the 5.5-volt driving force was also chosen based on the voltage used for the Tectonic actuators. Increasing or decreasing the magnitude of the orthogonal vibration may also improve perceptibility.

Most importantly, all our participants, without prior experience, immediately saw the usefulness and possible applications for our research. Without exception, all participants felt a vibration moving dynamically across the surface of the tablet. There is much still room to improve the distinctiveness of each targeted, localized vibration. Nevertheless, our data, both objective and subjective, are the first research results we have collected that demonstrate the possibility of improving localization detected by the fingertips.

## 7. Conclusions

The ability to induce high-fidelity perceptible haptics within a surface with the simplicity of four haptic actuators puts us closer to a process that can be realistically and reliably integrated into the manufacturing process. While traditional individual tactile actuation does provide accurate feedback, its complex manufacturin, and high number of components make it impractical to implement in current consumer technology.

It was found that the Nylon mesh and Teflon covering resulted in an impact on user performance that is statistically significant ($p < 0.05$). While the Silicone covering, having three times greater friction than Gorilla Glass, had little to no impact on decision time, the number of task repetitions, or error rate ($p > 0.05$). The inclusion of the lateral vibration modifier (60 Hz) successfully nearly doubled performance, as demonstrated for both the Nylon mesh and Teflon covering. We believe, due to these favorable results, that it would be worthwhile to expand the collection of materials studied as well as open the study up to more participants. The long-term goal is to find a combination of vibration

localization techniques combined with a compliant material that would lead to a result with repeatability with the introduction of outside distractions. Other aspects of the given materials will need to be further studied to better understand what material properties provide the overall best enhancement. This may include testing varying densities and thicknesses of a given material.

Logically we can assume that as the duration of each localized point decreases from 50 to 35 ms, the LVM would also need to be adaptively decreased from 15 to about 10 ms. That is to say that the LVM operating frequency should be increased from 60 Hz to about 100 Hz in order to consistently adapt to the length of duration for each localized vibration point. This would put the Lofelt L5 actuator outside of its resonance frequency, which may degrade performance. In which case it would be better to include actuators for the LVM that have a higher resonance frequency. To minimize undesirable side effects of the component-specific parameter, we can, for example, select the HAPTIC$^{TM}$ Reactor D [36], which operates at a resonance frequency of 160 Hz. Providing an LVM force that is equal to the range of ideal frequencies would likely involve operating multiple actuator technologies in sync and at varying operating voltages. While this leaves much to be studied in regards to the LVM, we believe that providing an LVM with dynamically changing frequency with an equalized frequency range is likely to provide better and more consistent results.

Vibration is a key component of handheld devices. The introduction of intermediate layers such as Teflon can improve the perception of vibrations over existing surfaces. How pleasant a material feels impacts how and how often a user chooses to interact with a device. The introduction of flexible displays makes the use of rigid Gorilla Glass surfaces problematic. A surface material such as Teflon could provide a comfortable material to users in such devices.

**Author Contributions:** Conceptualization, P.C. and G.E.; software, P.C.; validation, P.C.; formal analysis, P.C. and G.E.; investigation, P.C.; resources, R.R.; data curation, G.E.; writing—original draft preparation, P.C.; writing—review and editing, G.E. and R.R.; visualization, G.E.; supervision, G.E. and R.R; project administration, R.R.; funding acquisition, R.R. All authors have read and agreed to the published version of the manuscript.

**Funding:** This research was funded by Business Finland grant number 1316/31/2021.

**Institutional Review Board Statement:** Study did not require ethical approval.

**Informed Consent Statement:** Informed consent was obtained from all subjects involved in the study.

**Data Availability Statement:** The data are not publicly available due to participants' personal information.

**Conflicts of Interest:** The authors declare no conflict of interest.

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
