# Peer review of "The Impact of Different Overlay Materials on the Tactile Detection of Virtual Straight Lines"

_mti, doi:10.3390/mti7040035_

Round 1

Reviewer 1 Report

The paper presents a Study to improve the perception of haptic feedback Teflon using Nylon mesh and Silicone overlays tested in combination with lateral vibrations to study their impact on the tactile sense.

Some suggestions

The Abstract must be improved with some results and more description of the study.

The Introduction can have more examples about the State of the art and before the 1.1. can be presented the content of the paper.

The algorithms used in the data processing must be presented.

Future work can be detailed.

The Figure 2, 3 ..  can be noted with a) and b) -replace the Bottom and Top

Fig.8 can be replace left and right with a and b)

Author Response

The paper presents a Study to improve the perception of haptic feedback Teflon using Nylon mesh and Silicone overlays tested in combination with lateral vibrations to study their impact on the tactile sense.

Some suggestions

The Abstract must be improved with some results and more description of the study.

--The abstract has been modified for clarity. Summarized results have now been included.

The Introduction can have more examples about the State of the art and before the 1.1. can be presented the content of the paper.

--A paragraph has been added to the introduction to better expand on the current state of material technology and why it is of interest. Otherwise, discussion on most work in this area has already been discussed.

The algorithms used in the data processing must be presented.

--A paragraph has been added to the experimental design to better clarify how signal data is processed.

Future work can be detailed.

--future work has been briefly added to the conclusion.

The Figure 2, 3 ..  can be noted with a) and b) -replace the Bottom and Top

Fig.8 can be replace left and right with a and b)

-Changes to figures made.

Reviewer 2 Report

Regarding the paper titled “The impact of different overlay materials on the tactile detection of virtual straight lines” (paper ID: mit-2273699), I believe that an overall good job has been done reporting interesting content regarding quality and originality. The paper is well structured, and the English used is fine. However, some shortcomings related to the description of carried out the scientific activity and corresponding results are found, which should be solved to improve the quality and readability of the article. The main issues found in the manuscript are:

-        The authors should review the abstract to introduce better and summarize the carried-out work.

-        Relatively to the Introduction, the authors should better clarify the novelties and contributions of the proposed work.

-        At the end of the Introduction, the authors should briefly summarize the paper's structure.

-        The authors are suggested to review the whole manuscript to improve the English language and correct typos.

-        The authors should substitute some figures with high-quality versions (e.g., Figures 3a, 4, 5, 6).

-        The authors should revise the Conclusions to better summarize the main obtained results (also providing numerical value); besides, the future development of the carried out work could be introduced.

Author Response

Regarding the paper titled “The impact of different overlay materials on the tactile detection of virtual straight lines” (paper ID: mit-2273699), I believe that an overall good job has been done reporting interesting content regarding quality and originality. The paper is well structured, and the English used is fine. However, some shortcomings related to the description of carried out the scientific activity and corresponding results are found, which should be solved to improve the quality and readability of the article. The main issues found in the manuscript are:

-        The authors should review the abstract to introduce better and summarize the carried-out work.

The abstract has been modified for clarity. Summarized results have now been included.

-        Relatively to the Introduction, the authors should better clarify the novelties and contributions of the proposed work.

---The introduction has now been improved with the addition of key results. An additional paragraph to clarify the novelty and purpose of the study has been added. Description of the study has been expanded upon.

-        At the end of the Introduction, the authors should briefly summarize the paper's structure.

--A summary of the paper has been included at the end of the introduction.

-        The authors are suggested to review the whole manuscript to improve the English language and correct typos.

Manuscript has been reviewed and corrected where required.

-        The authors should substitute some figures with high-quality versions (e.g., Figures 3a, 4, 5, 6).

All images have been replaced with higher quality versions.

-        The authors should revise the Conclusions to better summarize the main obtained results (also providing numerical value); besides, the future development of the carried out work could be introduced.

Conclusion has been modified to better summarize results. Future development has been addressed.

Round 2

Reviewer 1 Report

The paper "The impact of different overlay materials on the tactile detection of virtual straight lines" can be published in the present form.